# Plasma Sphingoid Base Profiles of Patients Diagnosed with Intrinsic or Idiosyncratic Drug-induced Liver Injury

**DOI:** 10.3390/ijms24033013

**Published:** 2023-02-03

**Authors:** Zhibo Gai, Sophia L. Samodelov, Irina Alecu, Thorsten Hornemann, Jane I. Grove, Guruprasad P. Aithal, Michele Visentin, Gerd A. Kullak-Ublick

**Affiliations:** 1Department of Clinical Pharmacology and Toxicology, University Hospital Zurich, University of Zurich, 8006 Zurich, Switzerland; 2Experimental Center, Shandong University of Traditional Chinese Medicine, Jinan 250355, China; 3Institute of Clinical Chemistry, University Hospital Zürich, 8006 Zurich, Switzerland; 4NIHR Nottingham Biomedical Research Centre, Nottingham University Hospitals NHS Trust & University of Nottingham, Nottingham NG7 2UH, UK; 5Nottingham Digestive Diseases Centre, Translational Medical Sciences, School of Medicine, University of Nottingham, Nottingham NG7 2UH, UK; 6Mechanistic Safety, CMO & Patient Safety, Global Drug Development, Novartis Pharma, 4056 Basel, Switzerland

**Keywords:** acetaminophen, deoxysphingolipids, drug-induced liver injury, DILI, paracetamol, sphingolipids

## Abstract

Sphingolipids are exceptionally diverse, comprising hundreds of unique species. The bulk of circulating sphingolipids are synthesized in the liver, thereby plasma sphingolipid profiles represent reliable surrogates of hepatic sphingolipid metabolism and content. As changes in plasma sphingolipid content have been associated to exposure to drugs inducing hepatotoxicity both in vitro and in rodents, in the present study the translatability of the preclinical data was assessed by analyzing the plasma of patients with suspected drug-induced liver injury (DILI) and control subjects. DILI patients, whether intrinsic or idiosyncratic cases, had no alterations in total sphingoid base levels and profile composition compared to controls, whereby cardiovascular disease (CVD) was a confounding factor. Upon exclusion of CVD individuals, elevation of 1-deoxysphingosine (1-deoxySO) in the DILI group emerged. Notably, 1-deoxySO values did not correlate with ALT values. While 1-deoxySO was elevated in all DILI cases, only intrinsic DILI cases concomitantly displayed reduction of select shorter chain sphingoid bases. Significant perturbation of the sphingolipid metabolism observed in this small exploratory clinical study is discussed and put into context, in the consideration that sphingolipids might contribute to the onset and progression of DILI, and that circulating sphingoid bases may function as mechanistic markers to study DILI pathophysiology.

## 1. Introduction

Profiling plasma lipids, as a reflection of hepatic lipid content [1], has become a powerful translational approach to better understand the role of different lipid species in liver diseases [2], and has been exploited in the study of hepatocellular carcinoma [3,4] and nonalcoholic fatty liver disease [5,6]. A major contributor to the heterogeneity of the plasma lipidome, and to the informative potential thereof, are the sphingolipids, which constitute only about 4% of total plasma lipids but comprise hundreds of distinct species [7]. Sphingolipids can be classified in accordance to their complexity. The first level of complexity stems from the sphingoid bases, and their simple derivatives. Sphingoid bases are long-chain saturated (sphingosine, SO) and non-saturated (sphinganine, SA) amino alcohols that represent the basic structural constituents of sphingolipids (Figure 1). The second level of complexity comprises sphingoid bases that are N-acetylated with a second fatty acid (e.g., ceramides). Finally, more complex sphingolipids display an O-linked head group (e.g. sphingomyelins, glycosphingolipids) [8]. Sphingomyelins are the major sphingolipid species found in plasma, constituting ~95% of total plasma sphingolipids [7]. With respect to the sphingoid bases, the profile is more dynamic. C18-sphingosine (C18SO) is the most abundant species, representing ~60% of the total sphingoid base backbones. C18-sphinganine-diene (C18SA-diene), C18SA and C16SO represent 18%, 9% and 10%, respectively [9]. Other sphingoid bases are present to a minor extent [7]. Sphingoid bases are formed by the enzyme serine palmitoyltransferase (SPT) as the first and rate limiting step of the cellular sphingolipid de novo synthesis. The heterogeneity in the sphingoid base profile is related to the intrinsic polyspecificity of SPT [10]. From a biological standpoint, sphingolipids are essential as building elements of cellular membranes, as intra- and extracellular signaling modulators, and as substrates for metabolic pathways. Alterations in the plasma sphingolipid content and profile have been associated with metabolic syndrome and cardiovascular disease in both animals and patients [11,12,13,14,15,16].

Several animal and in vitro studies have observed altered sphingolipid levels and/or metabolism upon treatment with known hepatotoxic drugs. A study in rats has reported changes of plasma ceramides upon treatment with drugs inducing hepatic phospholipidosis, such as the tricyclic antidepressant clomipramine, and the antifungal ketoconazole [17]. Balb/c mice treated with triptolide, a hepatotoxic herbal medicine, revealed significant changes in sphingolipid metabolism at the transcriptional level, which influenced the hepatic and plasmatic sphingolipid profile [18]. Furthermore, the hepatotoxicity of Fumonisin B1 (FB1), which inhibits ceramide synthases, closely correlated with the accumulation of free sphinganine in mice [19,20], leading also to elevations in cytotoxic 1-deoxySA in vitro and in mice [21]. Substantial changes were measured in the cellular sphingolipid profile of human primary hepatocytes exposed to toxic extracellular concentrations of acetaminophen (APAP), a well-characterized hepatotoxic drug [22]. Taken together, these studies suggest that a link between sphingolipid metabolism and drug-induced liver injury (DILI) might exist. 

DILI is an adverse event that is often reversible; however, some patients can develop chronic liver injury and, in some cases, acute liver failure occurs. DILI is indeed the most frequent cause of acute liver failure in the U.S.A. and Europe [23,24]. Except for APAP-induced liver injury, which is the result of the intrahepatic accumulation of its reactive metabolite N-acetyl-p-benzoquinone imine (NAPQI), being dose-dependent and having an onset typically shortly after exposure (intrinsic DILI) [25], DILI is largely an idiosyncratic event that poses a true diagnostic challenge because it is dose-independent and may manifest only after a relatively long latency period after exposure to causative agents. Literature suggests that the frequency of idiosyncratic DILI varies strongly across drugs and drug classes, occurring in a European population-based study in 5 to 750 individuals out of every 100.000 exposed [26], underscoring genetic predisposition as a key risk factor. A large body of evidence suggests that idiosyncratic DILI is often immune-related, arguably triggered by a faulty adaptive immune response involving certain human leukocyte antigen (HLA) system variants. HLA polymorphisms associated to DILI have been reported for a number of drugs, including amoxicillin in combination with clavulanic acid [27,28], flucloxacillin [29], penicillin [30], terbinafine [31] and sulfomethazole/trimethoprim [32]. 

As studies in humans on the perturbations of sphingolipid metabolism caused by DILI are missing, we performed a simplified sphingoid base analysis of the plasma of patients with suspected DILI, comparing the findings with plasma profiles obtained from healthy volunteers. We found that the sphingolipid content and profile of the plasma of the DILI patients overlapped with that of healthy volunteers, and that the only significant change was in the level of 1-deoxysphingolipids. We then interrogated the circulating sphingolipidome of the DILI patients to explore the informative potential of circulating sphingolipids with respect to type, severity, and perpetrator drug.

## 2. Results

### 2.1. Clinical Outcome

Twenty-eight patients were referred to our center with the suspicion of DILI and included in the Swiss Drug-Induced Liver Injury Cohort Study. Based on the pattern of serum enzyme elevations at disease onset (R ratio: [(ALT/ULN) ÷ (ALP/ULN)]), 20 cases were characterized as suffering a hepatocellular injury (R > 5.0), six cases were defined as cholestatic (R < 2.0) and two cases as a mixed type of injury (R = 2.0–5.0). Six patients met the criteria of “nR Hy’s law” ([(ALT/ULN) ÷ (ALP/ULN)] > 5 and total bilirubin (TBL) > 2.5 mg/dL) [33], hence considered at a higher risk of progressing to acute liver failure, with one of these patients progressing to acute liver failure and receiving a liver transplant. One non-nR Hy’s Law case was fatal (advanced age and several comorbidities). In all patients, liver function parameters that were found elevated at the time of the diagnosis decreased after the discontinuation of the suspected DILI-causing medications (dechallenge), whereby in five cases the discontinuation date of the suspected medication was not known. In 11 cases, the liver enzymes did not normalize completely by the end of the follow-up, in 12 cases a restitutio ad integrum could be demonstrated. A rechallenge, i.e., re-administration of the suspected drugs, did not take place in any of the cases. After the suspected causative drugs were discontinued, a median of 3 days was necessary for the functional parameters to regress (between 0 and 48 days). For the 12 restitutio ad integrum cases, liver function normalized after an average of 36 days from the discontinuation of the suspected drugs (between 5 and 251 days).

Across all cases, the two main classes of perpetrator drugs were analgesics, mainly APAP, and antibiotics, primarily amoxicillin in combination with clavulanic acid. APAP was being taken by a total of 14 patients. Seven patients were within normal dose ranges (up to 4 g/24 h), where other substances being taken aside from APAP were identified as the suspected perpetrator drugs, resulting in 21 idiosyncratic DILI cases and seven high-dose APAP (intrinsic) DILI cases. In the latter patients, APAP could be assumed the sole perpetrator drug in four cases of intentional overdose, whereas for all seven cases, high doses of at least or more than a single ingestion of 24 g within 24 h were recorded, or high serum levels were detected (>380 µmol/L) in one case where the dose was unknown. The causal relationship between the suspected drugs and liver damage was evaluated using the Roussel Uclaf Causality Assessment Method (RUCAM) [34] in all cases of idiosyncratic DILI cases (not caused by high-dose APAP). Of 21 idiosyncratic DILI cases, the causality between drugs and liver damage was classified as “highly probable” (score > 9) in eight cases, in 12 cases as “probable” (score of 6–8) and in the remaining six cases as “possible” (score of 3–5). Referring to the treatment history of the 28 patients, 34 potentially hepatotoxic substances were identified.

### 2.2. Plasma Sphingolipid Levels in Healthy Volunteers and Patients Diagnosed with DILI

Thirteen sphingoid bases were measured in the plasma of the patients with DILI and the control subjects. Suspected intrinsic (APAP overdose) and idiosyncratic cases were grouped together in initial analyses due to the available case numbers and clinical descriptions. The calculated concentrations of the different sphingoid bases agreed well with previous analytical studies [35,36,37,38]. Firstly, we assessed the effect of the co-variants gender and age on plasma sphingolipid profile across both groups and found that neither seem to affect the plasma sphingolipid profile (Figure 2A,C). As alterations in the sphingolipid content and profile have been associated with cardiovascular disease (CVD) in both animal and human studies [11,12,13,14], the effect of underlying CVD on sphingolipid content and profile was also investigated. It can be seen that the total level of sphingolipids (Figure 2D) and the level of most of the sphingoid bases (Figure 2B) were comparable between the CVD and the non-CVD group. However, the level of circulating 1-deoxysphingosine (1-deoxySO) was significantly higher in the CVD group than in the non-CVD group (0.17 ± 0.02 vs. 0.10 ± 0.01 μmol/L) (Figure 2B,E) when comparing all cases and controls. 

Next, we compared the plasma sphingolipid profile of DILI and control groups. It can be seen that the level of total sphingolipids (Figure 3A) and of the individual sphingoid bases (Figure 3B) was comparable between the two groups. However, when the analysis was performed upon exclusion of the individuals with underlying CVD in both groups, it was found that the plasma level of 1-deoxysphingosine (1-deoxySO) was significantly elevated in the DILI group in comparison to the control group (0.13 ± 0.02 vs. 0.06 ± 0.02 μmol/L, *p* = 0.0045) (Figure 3C,D). Figure 3E shows that the level of 1-deoxySO in the plasma of patients with DILI did not correlate with the ALT values measured at the time of the diagnosis, suggesting that the higher level of circulating 1-deoxySO did not reflect the extent of the liver damage, as defined by the ALT value.

### 2.3. Sphingolipid Profile in DILI Subgroups

Among the thirteen sphingoid bases measured in plasma, the level of 1-deoxySO was the only significant difference between individuals with ongoing DILI and healthy volunteers. This difference could be detected only when comparing cases and controls without underlying CVD. Further analyses on the non-CVD groups were carried out based on the type or severity of DILI. Based on the R ratio, 11 hepatocellular and five cholestatic/mixed DILI were observed. In Figure 3A, the plasma level of 1-deoxySO was comparable between the two groups. Then, DILI cases were regrouped according to Hy’s law and non-Hy’s law cases, using the revised nR Hy’s law where DILI cases (i) displaying an R ratio > 5 ([(ALT or AST (whichever is highest) /ULN) ÷ ALP/ULN)] > 5) and (ii) serum TBL of greater than 2.5 mg/dL were considered severe (fulfilling nR Hy’s law criteria) [33,39]. Similar to the type of injury analysis, the plasma level of 1-deoxySO was comparable between the two groups (Figure 4B). Five out of 16 DILI cases without underlying CVD were caused by APAP intoxication. The level of 1-deoxySO in the APAP group was significantly higher than that in the control group (Figure 5A). Yet, most of the other sphingoid bases were underrepresented in the APAP group (Figure 5C) as reflected by the significantly lower level of total sphingolipids (Figure 5B). In particular, in Figure 4, it can be seen that the circulating level of the atypical sphingoid bases C16 and C17 were significantly reduced in the plasma of individuals who experienced APAP intoxication.

## 3. Discussion

The level of sphingolipids largely depends on the expression and activity of the serine palmitoyltransferase (SPT), which catalyzes the formation of the sphingoid base from acyl-CoA species, preferentially palmitoyl-CoA, and serine, and is considered the rate-limiting step in sphingolipid de novo synthesis [40]. SPT is an endoplasmic reticulum (ER) protein ubiquitously expressed in the body, and its expression level and activity have been shown to increase in response to several stress stimuli [40]. With respect to the liver, it has been shown that the expression and activity of SPT was significantly induced in rats treated with hexachlorobenzene, a chlorinated hydrocarbon associated with porphyria [41]. We have previously shown that the level of hepatic and circulating sphingolipids are increased in a diet-induced non-alcoholic steatohepatitis (NASH) mouse model [14], but not in the plasma of patients with non-alcoholic fatty liver disease (NAFLD), suggesting that sphingolipid metabolism adaptation to liver injury might differ between humans and rodents [14]. In line with these results, the level of total sphingolipids is not increased in the plasma of patients diagnosed with DILI in comparison to that measured in the plasma of healthy volunteers. 

1-deoxysphingolipids (mostly 1-deoxySO) are atypical sphingolipids formed when SPT metabolizes alanine instead of serine. 1-deoxysphingolipid levels have been found to be elevated in patients carrying SPT mutations, causing hereditary sensory autonomic neuropathy type 1 [42,43], and, while no studies specifically confirm elevations in patients with CVD, in metabolic disorders such as nondiabetic metabolic syndrome and type 2 diabetes [12,13,14,44]. 1-deoxysphingolipids are toxic metabolites that seem to target mitochondria, causing mitochondrial dysfunction and ER stress [45,46]. It is conceivable that the elevated 1-deoxysphingolipids content measured in the plasma of the individuals that experienced DILI might contribute to extensive mitochondrial damage and ER stress in hepatocytes. Being highly enriched in mitochondria to meet the high ATP demand to sustain the exceptional anabolic and catabolic activity that the liver must ensure to the body, hepatocytes must cope with a relatively high level of reactive oxygen species (ROS) that are normally generated during mitochondrial respiration [47]. An insult at the expense of the mitochondria can rapidly elevate intrahepatic ROS level to toxic concentrations that exacerbate the initial mitochondrial damage, endangering the whole cell. ER stress appears to be exceptionally high in DILI [48]. Moreover, the level of oxidation protein products, a typical sign of ER stress, has been shown to correlate well with the severity of the hepatic damage [49]. 

The sphingoid base profile does not appear to be informative with regard to the type and the severity of the injury. Conversely, when the DILI cases were grouped into APAP-induced (intrinsic) DILI and other (idiosyncratic) DILI, a pattern emerged. C16-sphinganine (C16SA), C16-sphingosine (C16SO) and C17-sphinganine (C17SO) were all consistently reduced in the plasma of individuals with intrinsic DILI but not in the idiosyncratic DILI group. C16SA and C17SA are atypical sphingoid bases, formed by reaction of SPT with acyl-CoAs with different carbon numbers than palmitoyl-CoA. Mammalian SPT is ubiquitously present as a tetramer formed by two dimers of the subunits SPTLC1 and SPTLC2. In some tissues, SPTLC1 predominantly dimerizes with a third subunit, SPTLC3, which has the highest affinity for the acyl-CoAs other than palmitoyl-CoA [50,51]. It has been shown that the expression levels of SPTLC1 and SPTLC2 are induced by acute ER stress in primary hepatocytes and HepG2 cells. Moreover, Sptlc2 was upregulated by tunicamycin, an ER stress inducer, in the livers of C57BL/6J wild-type mice [52]. In another study, it has been shown that SPT activity in the liver of Syrian hamsters and in HepG2 cells was elevated during inflammation [53]. Although the liver also expresses SPTLC3, data on the role of ER stress and inflammation on hepatic SPTLC3 expression are lacking [54]. It is conceivable that an increase in the SPTLC2:SPTLC3 induced by acute ER stress and/or inflammation would favor the synthesis of canonical (C18-, C19-sphingolipids) over atypical sphingolipids (C16-, C17-sphingolipids). ER stress is not specific to APAP-induced liver injury, although well known for this drug and likely secondary to extensive mitochondrial stress, and has been observed with several other DILI-causing drugs [55,56]. However, ALT values for the APAP-DILI patients included in this study were strongly elevated (all cases presented with ALT > 2300 U/liter), which may indicate advanced inflammatory processes and ER stress in comparison with the idiosyncratic DILI cases, and result in the observed alterations in sphingoid base profiles in these patients. As ALT levels do not correlate with severe clinical outcomes in acute or chronic liver diseases, the potential of select sphingoid bases functioning not only as biomarkers offering mechanistic insights into liver injury, but also serving alongside standard measures as independent conditional variables in the assessment of DILI, will warrant further studies in both intrinsic and idiosyncratic DILI cases.

## 4. Materials and Methods

### 4.1. Patient Selection

Between 2012 and 2015, patients presenting with possible drug-induced liver injury (DILI), as recorded by the pharmacovigilance service of the department of Clinical Pharmacology and Toxicology at the University Hospital Zurich, associated outpatient clinics, and GZO Spital Wetzikon, were enrolled into the Swiss Drug-Induced Liver Injury Cohort Study, giving informed consent for participation in the observational study. The study was conducted in accordance with the Declaration of Helsinki, and approved by the Swiss Cantonal Ethics Committee of Zürich (Kantonale Ethikkomission Zürich, SDILIC study ID KEK-ZH-Nr. 2012-0166). Cases were reviewed and potential causative agents determined by an adjudication committee consisting of a team of independent clinical experts, including at least one experienced pharmacologist. Differential diagnosis of DILI was made after careful assessment of patient and medication records, using the Roussel Uclaf Causality Assessment Method (RUCAM) for idiosyncratic DILI cases to assess causality, and APAP doses, serum APAP levels, and liver enzyme time course data to assess cases where intrinsic DILI was suspected. Blood samples were collected from patients on the day of trial inclusion/diagnosis of DILI and regularly during the course of clinical care. Sphingoid bases and liver laboratory parameters reported here were quantified in initial samplings obtained upon diagnosis of DILI and inclusion into the study. For sphingoid base quantification, blood was collected in EDTA tubes (BD Vacutainer system, BD Plymouth, Plymouth, UK), centrifuged at 2600g_av_ for 10 minutes, aliquoted, and immediately placed at −80 °C. Patient characteristics are provided in Table 1. Control patients without elevated liver enzymes or pre-existing diagnosed liver diseases were recruited within the same study, with blood samples being taken on the day of inclusion to the study. Inclusion and exclusion criteria for DILI and non-DILI control patients are shown in Appendix A. Suspected perpetrator drugs for DILI patients are shown in Appendix A. To supplement the 6 non-DILI control samples, 9 anonymized randomly selected biobanked samples collected from consenting normal healthy volunteers without pre-existing diagnosed CVD, liver disease, or diabetes, were included in the reported analyses as controls, for a total of n = 15 controls. Select demographic, clinical, and normalized sphingolipid data for all subjects are reported in Appendix A.

### 4.2. Classification According to Type, Severity and Mechanism of Injury 

The liver injury was categorized as hepatocellular, cholestatic or mixed based on the R ratio [(ALT/ULN) ÷ ALP/ULN)]. DILI cases with an R factor >5 were defined as hepatocellular injury, whereas those cases with an R factor <2 were considered of cholestatic nature. An R factor between two and five defined a mixed injury [57]. nR Hy’s law was used to identify severe cases of injury [33]. For classification of DILI cases into intrinsic versus idiosyncratic, a conservative classification was performed in which only cases where APAP intoxication was recorded were classified as intrinsic, and only cases where perpetrator drugs with existing indications of adverse liver events, as recorded in the LiverTox® database and other literature as cited throughout, were classified as “other”/idiosyncratic DILI [58].

### 4.3. Sphingolipid Analysis

Sample preparation was done as described previously [44]. Briefly, 100 ul plasma was extracted in 0.5 ml MetOH spiked with 200 pmol d7-Sphingosine and d7-sphinganiene (Avanti Polar Lipids Inc, Alabaster, AL, United States). After extraction, the lipids were hydrolyzed in 1N HCl (16 h, 65°C) to release the N-acyl chain and head groups. The resulting long chain bases were re-extracted in chloroform/water and dried as described previously [44]. For the analysis, the sphingoid bases were dissolved in 75 µL of sample buffer (56% MetOH, 34% EtOH, 10% H2O) and derivatized with 5 µL ο-phthalaldehyde (OPA) working solution (990 µL boric acid [3%] + 10 µL OPA [50 mg/mL in EtOH] + 0.5 µL 2-mercaptoethanol). The long chain bases were separated on a C18 column (Uptispere 120 Å, 5 µm, 125 × 2 mm, Interchim, Montluçon, France) coupled to a Transcend UPLC pump (Thermo Scientific, Reinach, BL, Switzerland). A binary solvent system was applied at a flow rate of 0.3 ml/min, with solvent A (1:1 MetOH/ammonium acetate (5 mM) in water) and solvent B (100% methanol). The column was equilibrated with a 1:1 mixture of mobile phase A and B followed by injection of 25 µL of the sample. Sphingoid bases were eluted by a linear gradient from A to B (25 min) followed by a regeneration phase with 100% B (7 min). Analysis was done on a Q-Exactive Orbitrap mass spectrometer (Thermo, Reinach, BL, Switzerland) in full scan mode using atmospheric pressure chemical ionization. The following MS settings were used: scan range of m/z 120-1200, mass resolution of 140000, automatic gain control target of 3.00E+06 and max injection time of 512 msec. Peaks were quantified using the Quan Browser Software (Thermo Scientific, Reinach, BL, Switzerland).

### 4.4. Statistical Analysis

Mann–Whitney U-Test and One-Way ANOVA for comparison of sphingolipid levels, and Pearson’s correlation analyses between ALT and sphingolipid levels were performed with GraphPad Prism (version 9.0 for Windows, GraphPad Software). Volcano plots were plotted using MetaboAnalyst (version 5.0). 

## 5. Conclusions

Despite all the limitations that are intrinsic of an observational study on a small number of patients and controls, including confounding factors and the impossibility to demonstrate causality, our data indicate the clinical relevance of drug-induced perturbations of the sphingolipid metabolism observed in vitro and in animals. While total sphingoid base levels were not altered across patient and control groups, 1-deoxySO was elevated in DILI patients, after removal of underlying CVD as a confounding factor. In addition, shorter chain sphingoid bases C16SO, C16SA, and C17SO were specifically and exclusively reduced in patients with intrinsic DILI (caused by APAP overdose). The role of these species and significance of alterations in the levels thereof in the onset, progression, and clinical outcome of DILI cannot be determined from this small study. Nonetheless, our results warrant further larger scale clinical investigations, perhaps on more informative global lipidomics analyses into (i) how alteration of lipid metabolism can provide new mechanistic insights into hepatotoxicity and their role in it, and (ii) whether specific lipid species can be used as biomarkers in the delineation of the type of DILI or assessment of severity of injury in the frame of both drug development and pharmacovigilance.

## Figures and Tables

**Figure 1 ijms-24-03013-f001:**
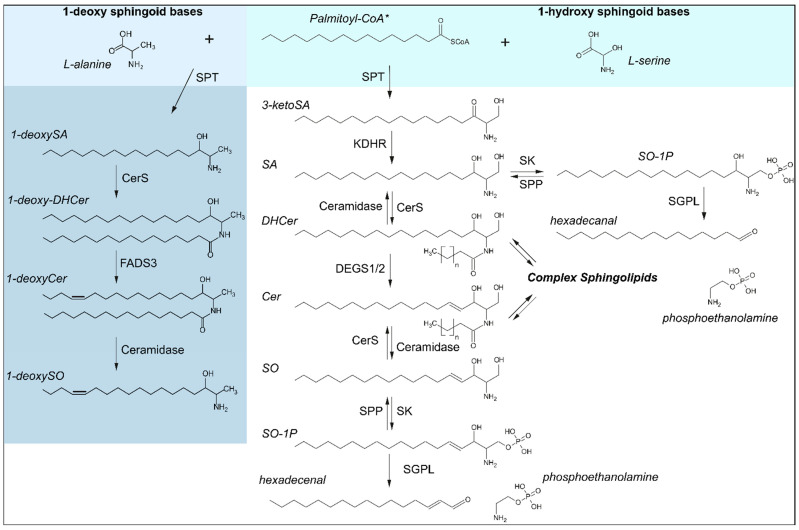
Sphingoid base biosynthesis. Synthesis of canonical (1-hydroxy) sphingoid bases is catalyzed over the condensation of acyl-CoAs, shown here with the preferred substrate* palmitoyl-CoA (C16:0), with L-serine by serine palmitoyltransferase (SPT) as the rate limiting step in de novo synthesis. 1-deoxy sphingoid bases are likewise synthesized by SPT, using the alternative substrate L-alanine. Enzymes involved are as follows: CerS: ceramide synthases, DEGS1/2: Dihydroceramide desaturases 1/2, FADS3: fatty acid desaturase type 3, KDHR: 3-ketoreductase, SGPL: sphingosine-1-phosphate lyase, SK: sphingosine kinase, SPP: S1P phosphatase, SPT: serine palmitoyltransferase. Substrates and products are abbreviated as follows: Cer: ceramide, DHCer: dihydroceramide, SA: sphinganine, SO: sphingosine, SO-1P: sphingosine-1-phosphate.

**Figure 2 ijms-24-03013-f002:**
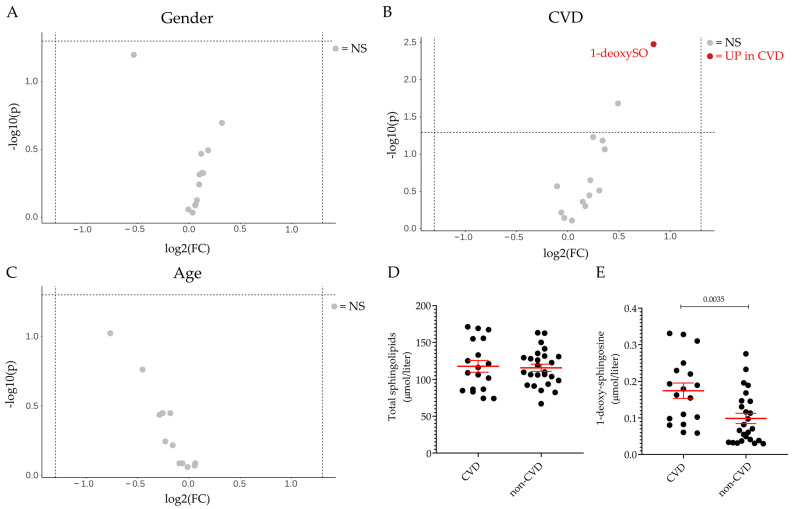
Gender, age and CVD effect on plasma sphingolipid profile. Volcano plot of the 13 sphingoid base species identified by LC–MS/MS according to gender (**A**), pre-existing cardiovascular disease (CVD) in individuals irrespective of DILI diagnosis (**B**), and age (≤50 vs. >50) (**C**). The fold-change of concentration is reported on the x-axis, the significance (*p*-value) on the y-axis. The vertical and horizontal dotted lines show the cut-off of fold-change = ±1.5, and of *p*-value = 0.05, respectively. Plasma level of total sphingolipids (**D**) and 1-deoxy-sphingosine (1-deoxySO) (**E**) in individuals with pre-existing cardiovascular disease (CVD group) and in those without CVD (non-CVD group). Data are expressed as mean ± SEM. The comparison of the means was performed by unpaired Student *t*-test. In scatter plot representation, each dot represents one individual sample. NS: not significant, UP: significantly increased.

**Figure 3 ijms-24-03013-f003:**
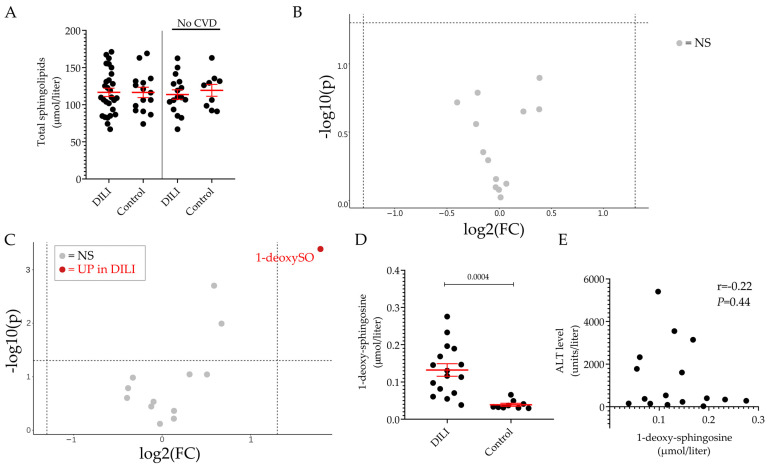
Plasmatic sphingolipid profile of individuals DILI patients and healthy volunteers. Total sphingolipid levels in DILI and control groups. Results are expressed as the mean ± SEM, where each dot represents one individual sample. Comparison of the means was performed by unpaired t-test (**A**). Volcano plot of the 13 sphingoid base species identified by LC–MS/MS between DILI and control groups (**B**) and between the non-CVD respective subgroups (**C**). The fold-change of concentration is reported on the x-axis, the significance (*p*-value) on the y-axis. The vertical and horizontal dotted lines show the cut-off of fold-change = ±1.5, and of *p*-value = 0.05, respectively. Plasma level of 1-deoxy-sphingosine (1-deoxySO) in non-CVD DILI and control groups (**D**). Pearson’s correlation analysis between ALT values and 1-deoxysphingosine in the non-CVD DILI group (**E**). NS: not significant, UP: significantly increased.

**Figure 4 ijms-24-03013-f004:**
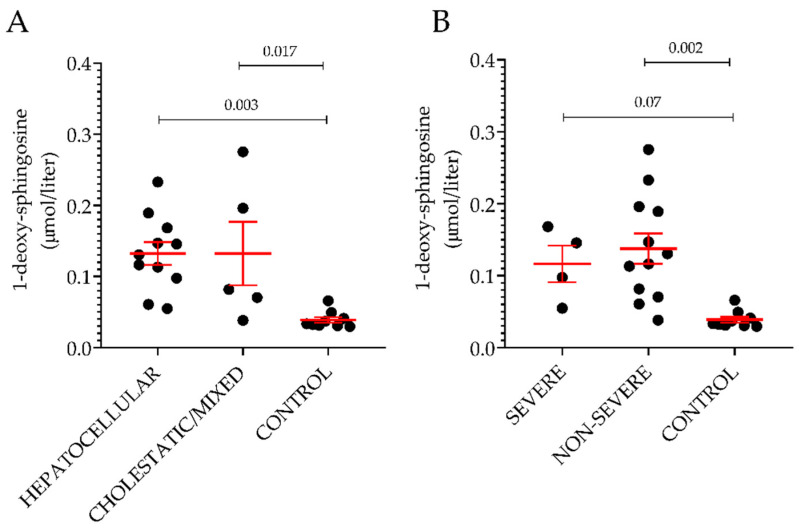
Plasma level of 1-deoxysphingosine according to type and severity of the injury. Classification was done based on the R ratio (**A**) or on the nR Hy’s law (**B**). Results are expressed as the mean ± SEM, where each dot represents one individual sample. Comparison of the means was performed by unpaired *t*-test.

**Figure 5 ijms-24-03013-f005:**
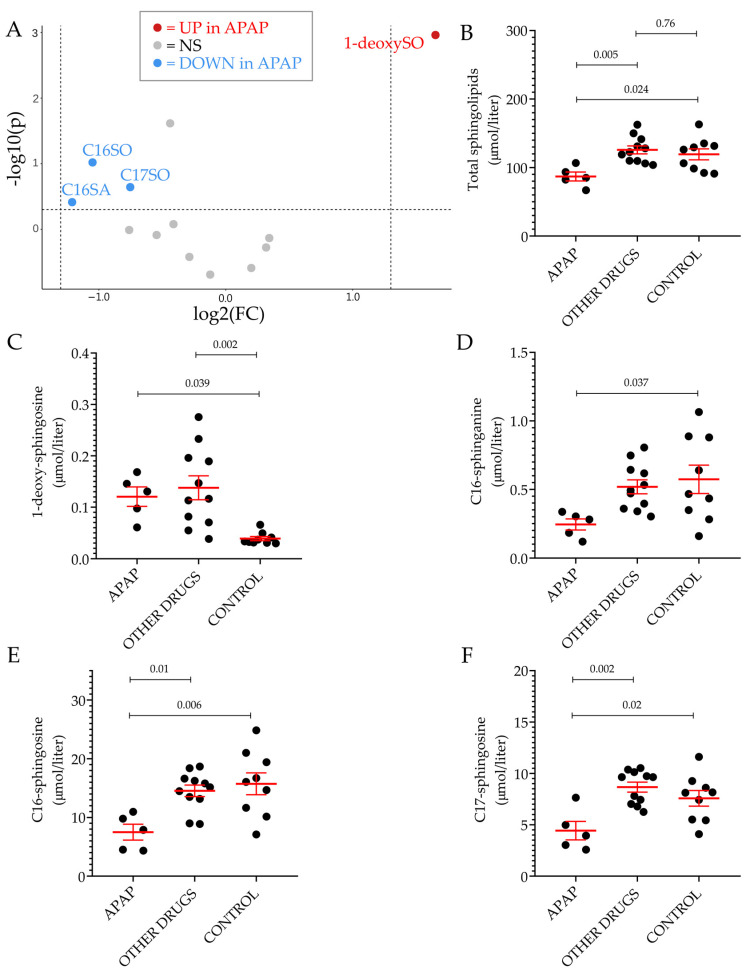
Plasmatic sphingolipid profile of patients with acetaminophen intoxication. Volcano plot of the 13 sphingoid base species identified by LC–MS/MS between individuals who experience APAP intoxication and the control group, excluding patients and controls with underlying CVD (**A**). The fold-change of concentration is reported on the x-axis, the significance (*p*-value) on the y-axis. The vertical and horizontal dotted lines show the cut-off of fold-change = ±1.5, and of *p*-value = 0.05, respectively. Scatter dot plot of total sphingolipid (**B**), 1-deoxysphingosine (**C**), C16-sphinganine (**D**), C16-sphingosine (**E**) and C17-sphingosine (**F**) levels in the indicated groups. In all scatter dot plots, results are expressed as the mean ± SEM, where each dot represents one individual sample. Comparisons of the means was performed by one-way analysis of variance (ANOVA) followed by Tukey’s multiple comparisons test. NS: not significant, UP: significantly increased. DOWN: significantly decreased.

**Table 1 ijms-24-03013-t001:** Demographics and patient characteristics.

		DILI (n = 28)	non-DILI (n = 15)
Parameters	Category	N (%)	
Age	≤55 >55	15 (54) 13 (47)	10 (67) 5 (33)
Gender	Male Female	17 (59) 12 (41)	7 (47) 8 (53)
Type of DILI	Hepatocellular Cholestatic Mixed	20 (69) 6 (21) 3 (10)	- - -
Severity of injury *	Severe Non-severe	8 (28) 21 (72)	- -
Steatosis	Yes No	4 (14) 25 (86)	- -
T2D	Yes No NA	4 (14) 19 (68) 5 (18)	1 (7) 14 (93) -
CVD	Yes No	12 (43) 16 (57)	6 (40) 9 (60)

* According to nR Hy’s law: increased alanine aminotransferase (ALT) or aspartate transaminase (AST) levels equal to or greater than 3× the upper limit of normal (ULN) and total bilirubin (TBL) levels greater than 2× the ULN. DILI: drug-induced liver injury, T2D: type II diabetes, CVD: cardiovascular disease.

## Data Availability

Additional patient data are available on request from the corresponding author.

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
