# Peer review of "Plasma Sphingoid Base Profiles of Patients Diagnosed with Intrinsic or Idiosyncratic Drug-induced Liver Injury"

_ijms, 2023, doi:10.3390/ijms24033013_

Round 1

Reviewer 1 Report

 Title: Plasma sphingolipid profile of patients diagnosed with drug-2 induced liver injury

The present manuscript (ijms-2106853) entitled “Plasma sphingolipid profile of patients diagnosed with drug-2 induced liver injury” reported by Z. Gai  et.al., was found to be suitable and acceptable to publish in the IJMS journal (Special issue: Molecular Mechanisms of Hepatotoxicity) with the following minor revisions.

1. The introduction part is lengthier and now a days, “profiling plasma lipids” definition / primary information is not need in an elaborated manner in the introduction part.

2. Contribution from recent years like 2021 or 2022 in the reference part will increase the beauty of the manuscript. Out of 52 references, only 3-4 were belongs to these years.

3. Although conclusion based points were present in the abstract, but generally, summary or conclusion based points were expected by a reader. Hence this can be included in the manuscript.

Reviewer 2 Report

The manuscript entitled “Plasma sphingolipid profile of patients diagnosed with drug induced liver injury”.

The researcher was highlighted in this study to investigate the profile of plasma sphingolipid of patients diagnosed with DILI. The authors are very impactful described all data in the manuscript which is easily understandable.

Researcher described the results of study in well structured manner in this study stepwise step this makes research more impactful.

Update introductory part with recent references.

Reference number 12 and 34 not complete check it.

Use uniform name or abbreviations throughout the manuscript.

Overall, good work done by researcher. The manuscript will be accepted after minor revision.
